# Sublingual Atropine Administration as a Tool to Decrease Salivary Glands’ PSMA-Ligand Uptake: A Preclinical Proof of Concept Study Using [^68^Ga]Ga-PSMA-11

**DOI:** 10.3390/pharmaceutics14061276

**Published:** 2022-06-16

**Authors:** Vincent Nail, Béatrice Louis, Anaïs Moyon, Adrien Chabert, Laure Balasse, Samantha Fernandez, Guillaume Hache, Philippe Garrigue, David Taïeb, Benjamin Guillet

**Affiliations:** 1Centre Européen de Recherche en Imagerie Médicale (CERIMED), Aix-Marseille University, CNRS, 13005 Marseille, France; beatrice.louis@etu.univ-amu.fr (B.L.); anais.moyon@univ-amu.fr (A.M.); adrien.chabert@univ-amu.fr (A.C.); laure.balasse@univ-amu.fr (L.B.); samantha.fernandez@univ-amu.fr (S.F.); guillaume.hache@univ-amu.fr (G.H.); philippe.garrigue@univ-amu.fr (P.G.); david.taieb@ap-hm.fr (D.T.); benjamin.guillet@univ-amu.fr (B.G.); 2Radiopharmacy Department, Assistance Publique-Hôpitaux de Marseille, CHU La Timone, CHU Nord, 13005 Marseille, France; 3Centre de Recherche en Cardiovasculaire et Nutrition (C2VN), Aix-Marseille University, INSERM, INRAE, 13005 Marseille, France; 4Nuclear Medicine Department, Assistance Publique-Hôpitaux de Marseille, CHU La Timone, CHU Nord, 13005 Marseille, France

**Keywords:** PSMA, prostate-specific membrane antigen, salivary glands, atropine

## Abstract

Prostate Specific Membrane Antigen (PSMA)-directed radionuclide therapy has gained an important role in the management of advanced castration-resistant prostate cancer. Although extremely promising, the prolongation in survival and amelioration of disease-related symptoms must be balanced against the direct toxicities of the treatment. Xerostomia is amongst the most common and debilitating of these, particularly when using an alpha emitter. It is therefore of main importance to develop new preventive strategies. This preclinical study has evaluated the effect of α-adrenergic and anticholinergic drugs on [^99m^Tc]TcO_4_^−^ Single Photon Emission Computed Tomography/Computed Tomography (SPECT/CT) and [^68^Ga]Ga-PSMA-11 Positron Emission Tomography (PET/CT). Methods: The effects of phenylephrine, scopolamine, atropine, and ipratropium on salivary glands uptake were evaluated in non-tumor-bearing mice by [^99m^Tc]TcO_4_^−^ microSPECT/CT. The most efficient identified strategy was evaluated in non-tumor-bearing and xenografted mice by [^68^Ga]Ga-PSMA-11 PET/CT. Results: Scopolamine and atropine showed a significant decrease in the parotid glands’ uptake on SPECT/CT whereas phenylephrine and ipratropium failed. Atropine premedication (sublingual route), which was the most effective strategy, also showed a drastic decrease of [^68^Ga]Ga-PSMA-11 salivary glands’ uptake in both non-tumor-bearing mice (−51.6% for the parotids, *p* < 0.0001) and human prostate adenocarcinoma xenografted mice (−26.8% for the parotids, *p* < 0.0001). Conclusion: Premedication with a local administration of atropine could represent a simple, safe, and efficient approach for reducing salivary glands’ uptake.

## 1. Introduction

The recent positive results of the VISION trial (NCT03511664) for both primary endpoints (OS and rPFS) have given a great impetus towards the widespread use of PSMA-targeted radioligand therapy in advanced castration-resistant prostate cancer (CRPC) following docetaxel therapy [1]. Introduction of [^177^Lu]Lu-PSMA therapy earlier in the course of the disease is currently being evaluated in the setting of clinical trials. The substitution of a beta emitter (such as [^177^Lu]Lu) by an alpha emitter ([^225^Ac]Ac) has also shown very promising results that however would need to be balanced against the direct toxicities, especially on the salivary glands [2]. Various attempts were studied to prevent xerostomia after radioligand therapy but have remained largely unsuccessful [3]. Therefore, innovative strategies are required to prevent such debilitating side effects.

The PSMA receptor is a type-II transmembrane glycoprotein that acts as a carboxypeptidase activity. PSMA is expressed on prostate epithelial cells and is upregulated in prostate cancer. Overall, 90% of prostate cancers are positive for PSMA followed by positron emission tomography (PET), regardless of their phenotype [4]. Despite the low-to-moderate staining intensity of PSMA on salivary glands, the uptake of radiolabeled small-molecule-based PSMA ligands is high, suggesting a predominant non-specific mechanism for tracer accumulation. This is in agreement with a low uptake in the salivary glands on PET using radiolabeled anti-PSMA antibodies [5]. Competitive displacement strategies have been studied showing a reduction of salivary gland uptake and tumor uptake of PSMA-based radiotracers [6,7,8]. In parallel, non-specific approaches have been evaluated such as local cooling [9,10,11], lemon juice [12], vitamins [13,14], botulinum toxin [15], monosodium glutamate [16,17], or polyglutamate [18,19]. Local cooling with ice packs placed over the parotid glands with the intention to reduce perfusion is the easiest method and attempts to promote a modest and transient reduction of radioligand.

Physiological salivary function is regulated by both sympathetic and parasympathetic neurons [20]. The sympathetic nervous system modulates the salivary protein secretion via the α1 and β1 adrenoceptors whereas the parasympathetic nervous system regulates the aqueous and electrolyte secretions mainly via the muscarinic Type 3 receptor [21]. In parallel, both systems could act on peripheral blood vessels and vascular wall contractility [22]. Based on these considerations, the aim of our study was to evaluate the impact of pharmacological modulation of the autonomic innervation on the salivary glands. A screening test was initially performed by the assessment of [^99m^Tc]TcO_4_^−^ parotid uptake on Single Photon Emission Computed Tomography/Computed Tomography (SPECT/CT) following the administration of commercially available alpha-adrenergic (phenylephrine) or anticholinergic drugs (atropine, scopolamine and ipratropium). The lead strategy was then evaluated on [^68^Ga]Ga-PSMA-11 PET/CT in both healthy and human prostate adenocarcinoma-xenografted mice.

## 2. Materials and Methods

### 2.1. Radiopharmaceuticals

#### 2.1.1. [^99m^Tc]Pertechnetate

Sodium [^99m^Tc]pertechnetate solution (Na^+^, [^99m^Tc]TcO_4_^−^) was eluted with a physiological isotonic solution (NaCl, 0.9% *m/v*) from a commercial [^99^Mo]Mo/[^99m^Tc]Tc generator (Tekcis^®^, Curium, Saclay, France). After elution, radiochemical purity (RCP) was assessed using radio-thin layer chromatography using a radio-chromatograph (miniGITA^®^, Elysia-Raytest, Liege, Belgium), with cellulose filter paper (Whatman #5, Cytiva, Velizy-Villacoublay, France) as the stationary phase, and acetone as the mobile phase.

#### 2.1.2. [^68^Ga]Ga-PSMA-11

All chemicals and solvents were obtained from commercial suppliers and used without further purification. PSMA-11 precursor was purchased from ABX Advanced Compounds (Radeberg, Germany) and was diluted in a sodium acetate buffer solution (0.8 mol L^−1^, pH = 4.5). PSMA-11 (2.5 µg, 250 µL) was added to Gallium-68 chloride ([^68^Ga]GaCl_3_, 100–200 MBq, 1 mL), which was previously eluted from a commercial [^68^Ge]Ge/[^68^Ga]Ga generator (Galliapharm, Eckert&Ziegler, Berlin, Germany). The radiolabeling was achieved after a 10-min incubation at 95 °C. Then, RCP was controlled by radio-thin layer chromatography (radio-TLC) using instant Thin Layer Chromatography–Silica Gel (iTLC-SG) paper (Agilent, Les Ulis, France) as the stationary phase, and a 1 mol L^−1^ methanol/ammonium acetate solution (1:1; *v/v*) as the mobile phase.

### 2.2. Cell Line

The human prostate adenocarcinoma LNCaP cells were gratefully obtained from Dr. P. Rocchi (Centre de Recherche en Cancérologie de Marseille, Aix-Marseille University, Marseille, France). LNCaP cells were cultivated in RPMI 1640 medium (Thermo-Fisher, Waltham, MA, USA) supplemented with 10% fetal bovine serum (Eurobio Scientific, Courtaboeuf, France) and 0.1% penicillin-streptomycin (Fisher Scientific, Illkrich, France). Fifty million per milliliter of trypsinized LNCaP cells at a passage number between 5 and 10 were resuspended in a mixture of half culture media and half Matrigel matrix (Corning, New York, NY, USA).

### 2.3. Animals

Eight-week-old BALB/c mice were purchased from Janvier Labs (Le Genest-Saint-Isle, France). Eight-week-old male BALB/c nude mice (Janvier Labs; Le Genest-Saint-Isle, France) were injected subcutaneously into the flank with five million of LNCaP cells under 200 µL. All mice were housed in enriched cages, placed in a room with controlled temperature and hygrometry with daily monitoring, and had access to food and drinking water ad libitum. All experiments were performed by experimented and authorized operators and were conducted in accordance with the Aix-Marseille University institutional animal care and use committee (CE14, Aix-Marseille University) and the French Ministry of Research (project authorization #32157 on 11 July 2021) according to the European Union directive 2010/63/EU and the recommendations of the Helsinki declaration. Mice (20–25 g) with a tumor size of 350.5 ± 200.4 mm^3^ were used for experiments (3–6 weeks after inoculation).

### 2.4. Pharmacological Modulation

Mice were treated by different drugs, routes of administration, and doses as described in Table 1. Anticholinergic drugs (atropine, scopolamine, and ipratropium) and an α-adrenergic drug (phenylephrine) were evaluated. Depending on the used molecules, the intraperitoneal (ip) or sublingual (sl) routes were assessed. Intraperitoneal injection was performed 15 min before the radiotracer administration. Sublingual administration consisted of placing a compress impregnated with 50 µL of the corresponding drug in the oral cavity 15 min before the radiotracer administration, which was renewed at 15 min for a total impregnation duration of 30 min.

### 2.5. SPECT/CT Imaging

Mice were anesthetized with isoflurane (induction at 4.0%, maintenance at 1.5%, Isovet, Osalia, Paris, France) and injected intraperitoneally with 15.1 ± 2.0 MBq/0.2 mL [^99m^Tc]TcO_4_^−^. Mice were imaged on a NanoSPECT/CT camera (Mediso, Budapest, Hungary). The NanoSPECT/CT camera is annually calibrated by the manufacturer. Additionally, an internal calibration is performed monthly.

MicroSPECT/CT images were acquired 45 min after the administration of the radiotracer. The SPECT scan acquisition lasted 20 min. SPECT parameters were set up as follows: 80 projections per 360° (20 projections per detector) with 18° steps (60s/projection); picture size: 256 × 256; zoom factor: 1.14; pixel size: 1.00 mm^2^. Reconstruction was performed using HiSPEC software v. 1.4.3049 (Scivis GmbH, Göttingen, Germany) and SPECT signal quantifications on manually drawn regions of interest (ROIs) were performed using InVivoScope software v.3.5 (InVicro, Boston, MA, USA). To reduce bias due to the pertechnetate accumulation in thyroids and its potential signal spillover, only the uptake in parotid glands was quantified. Results are expressed as mean ± SD percentage of the injected dose per mm^3^ (%ID/mm^3^).

### 2.6. PET/CT Imaging

Mice were anesthetized with isoflurane (induction at 4.0%, maintenance at 1.5%, Isovet, Osalia, Paris, France) and injected intraperitoneally with 5.7 ± 0.5 MBq/0.2 mL [^68^Ga]Ga-PSMA-11. Mice were imaged on a NanoScan PET/CT camera (Mediso, Budapest, Hungary). The NanoScan PET/CT camera is annually calibrated by the manufacturer. Additionally, an internal calibration is performed monthly.

A 20 min static mode acquisition was performed 90 min after injection of [^68^Ga]Ga-PSMA-11. The default PET protocol has been used as: voxel size: 0.4 mm; numbers of iterations: 4, coincidence: 1–5 Field of View (FOV) image size: 169 × 170 × 241). Reconstruction (CT attenuation corrected) was performed using Nucline software (Mediso, Budapest, Hungary) and PET signal quantifications on manually drawn ROIs were performed with VivoQuant software v.3.5 (InVicro, Boston, MA, USA). Results are expressed as mean ± SD %ID/mm^3^.

### 2.7. Statistics

The statistical analysis of parotid gland SPECT signals was assessed by a paired t-test. Salivary glands and tumor PET signals were compared using a two-way ANOVA followed by a Sidak’s multiple comparisons test. Data were normalized to control data and expressed as mean (%) ± standard error of the mean (SEM). Statistical analyses were performed with Prism v.9.0 software (GraphPad, San Diego, CA, USA). Statistical significance was defined as *p* ≤ 0.05.

## 3. Results

### 3.1. Screening of Drugs Affecting the Autonomic Innervation of Salivary Glands by [^99m^Tc]TcO_4_^−^ SPECT/CT

At 45 min post-injection (p.i.), the administration of an α-adrenergic agonist, phenylephrine by intraperitoneal route, did not reduce significantly [^99m^Tc]TcO_4_^−^ uptake in the parotids when compared to the control condition (percentage variation (PV) = −16.7%; *p* = 0.20; *n* = 3) (Figure 1A,F). Scopolamine (5 mg kg^−1^, ip route) significantly decreased the [^99m^Tc]TcO_4_^−^ uptake in the parotids by −28.7% when compared to control condition (*p* < 0.0058; *n* = 6) (Figure 1B,G). Atropine (15 mg kg^−1^, ip route) allowed us to achieve a better effect than the other compounds on the parotids’ uptake when compared to the control condition (PV = −34.9%; *p* = 0.039; *n* = 3) (Figure 1C,H). Local administration using sublingual route of atropine at 15 mg mL^−1^ induced the most important and significant reduction of [^99m^Tc]TcO_4_^−^ parotids’ uptake when compared to the control condition (PV = −41.6%; *p* < 0.0160; *n* = 6) (Figure 1D,I). At 90 min p.i., the uptake was still drastically decreased by 37.3% (*p* < 0.010; *n* = 6). Finally, the sublingual administration of ipratropium at 4 mg kg^−1^ did not show any effect on the parotid gland uptake (PV = +4.4; *p* = 0.55; *n* = 3) (Figure 1E,J). SPECT quantification data are summarized in Table 2.

### 3.2. Evaluation of the Effect of Sublingual Atropine on [^68^Ga]Ga-PSMA-11 Salivary Gland Uptake in Healthy Mice

In agreement with the study design, the most effective strategy with [^99m^Tc]TcO_4_^−^ SPECT/CT (15 mg mL^−1^ atropine by sublingual route) was then evaluated with [^68^Ga]Ga-PSMA-11 PET/CT. At 90 min p.i., the uptake significantly decreased by 50% in the parotid glands (atropine: PV = −51.6%; *p* < 0.0001; *n* = 7) and submandibular glands (atropine: PV = −48.5%; *p* = 0.0002; *n* = 7) (Figure 2A,B). PET quantification data are summarized in Table 3.

### 3.3. Impact of Sublingual Atropine [^68^Ga]Ga-PSMA-11 Uptake on Salivary Glands and Prostate Cancer in LNCaP Xenografted Mice

A 30% decrease was also observed in the salivary glands of LNCaP xenografted mice (parotids: PV: −26.8%, *p* < 0.0001; *n* = 9; submandibular glands: PV: −37.6%; *p* < 0.0001; *n* = 9). The tumor [^68^Ga]Ga-PSMA-11 PET signal was increased by 18% following sublingual atropine when compared to control (*p* = 0.012; *n* = 9) (Figure 3A,B). PET quantification data are summarized in the Table 4.

## 4. Discussion

PSMA-targeted radioligand therapy has shown a great potential in the treatment of advanced castration-resistant prostate cancer (CRPC). As for other radiopharmaceuticals, the major salivary glands demonstrate high levels of uptake and are therefore regarded as organs-at-risk in the setting of radionuclide therapy. Despite very low incidence of high-grade dry mouth observed in the VISION trial, this side effects must be recognized and monitored in the later phases. This is even more important in the application of PSMA-targeted alpha therapy, where salivary gland toxicity is currently the dose-limiting side effect.

Overall, local cooling with ice packs placed over the parotid glands with the intention to reduce perfusion is the easiest method and attempts to achieve a modest and transient reduction of radioligand (from 30 min pre-infusion through 2 h post-infusion of radiopharmaceuticals). Its contribution on PSMA radiotracers uptake is still unclear and debated. Based on [^68^Ga]Ga-PSMA-11, Van Kalmthout et al. showed that Standard Uptake Values (SUV), SUVmax and SUVpeak, were decreased by 10–15% in the cooled glands (*n* = 44) when compared to the control group (*n* = 45) [10]. In comparison, Yilmaz et al., evaluating 19 patients with [^177^Lu]Lu-PSMA-617, did not find any difference between cooled and contralateral (control) salivary glands [11]. Moreover, external cooling has to be investigated for PSMA-targeted alpha-therapy. 

Targeting the specific bindings to the salivary glands using non-radiolabeled PSMA inhibitors was also an early lead to attempt to manage xerostomia for targeted-PSMA radiotherapy. Competitive blocking strategy studies using 2-PMPA, TrisPOC-2-PMPA, or PSMA-11 showed significant reductions of tracer uptake in the salivary glands (and the kidneys). However, this reduction was followed by a decrease of tumor uptake which might hamper therapeutic efficacy [6,7,8,23].

Among these multiple strategies, acting on the input function of radiotracers, by reducing salivary gland perfusion, appeared as the most-used strategy to reduce and manage potential adverse effects with a limited impact on tumor uptake. This strategy is routinely used, by applying cold to reduce alopecia after standard chemotherapies, thus, decreasing the perfusion to the salivary glands during PSMA radioligand therapy [10]. The purpose of this present study was to evaluate various approval molecules to reduce the perfusion of the salivary glands. This work shows that anticholinergic drugs (ip route) were the most effective approaches for reducing perfusion, characterized by a lower [^99m^Tc]TcO_4_^−^ parotid gland uptake (respectively by 29 and 35% for 5 mg kg^−1^ of scopolamine and 15 mg kg^−1^ of atropine). The intraperitoneal administration of radiotracers and drugs is interesting in this context due its delayed systemic absorption in comparison to bolus intravenous injection, mimicking clinical infusion [21]. In order to apply this strategy to humans, the sublingual administration of atropine (15 mg mL^−1^) was evaluated and enabled a similar effect compared to i.p experiments, with a reduction of 36% of parotid gland uptake at 45 min with a prolonged effect.

The results were interestingly confirmed in [^68^Ga]Ga-PSMA-11 PET experiments, with a reduction of 50% of [^68^Ga]Ga-PSMA-11 uptake in the parotids. Finally, and very interestingly, this strategy did not inhibit tumor uptake in xenografted mice and was even increased by 18%, probably due to an increase of radiotracer bioavailability.

Soon, studying [^177^Lu]Lu or [^225^Ac]Ac PSMA-targeted radiotracers with atropine sublingually administrated must be explored to evaluate their biodistribution and impact on salivary glands uptake.

To our knowledge, this is the first time that atropine has been studied with regards to decreasing PSMA radiotracer uptake in the salivary glands. Recently, the muscarinic inhibition of salivary glands using glycopyrronium bromide did not show any effect on the salivary gland uptake neither for the PSMA radiotracers nor for radioiodine [24]. This study trends to our results obtained using ipratropium bromide (sl) as control.

Atropine is an anticholinergic molecule that reduces gastric, salivary, bronchial, and sweat secretions by competitively inhibiting the muscarinic Type 3 receptor. Atropine significantly reduces the rate of saliva production and creates a vasoconstriction of afferent vessels [22,23]. The vasoconstriction produced using anticholinergic molecules seems to be more intense and sustained in comparison with ice-pack strategies, which could explain the higher efficiency of anticholinergic premedication. Atropine is an old and well-known molecule, and is routinely used via intravenous administration as a preoperative medication for the reduction of salivary and bronchial secretions or as eye drops for inducing mydriasis for diagnostic purposes. Despite the absence of an adapted galenic form and robust safety data using this route, sublingual administration is regularly used off-label, is documented for the treatment of drooling even in children, and is still studied in clinical trials [24,25,26,27,28,29]. As used for the management of sialorrhea, one or two drops (0.5 to 1 mg) of 1% atropine eye drops every 4 to 6 h (<10 mg/day) are described as being tolerable with low frequency, and with reversible side effects [27,30]. This cheap molecule has a quick start of action, with effects observable after 15 to 30 min [31]. Besides atropine (sl), the transdermal route using scopolamine could be an interesting alternative, using an approved medication for human use that has similar expected results to atropine (sl) [32]. Its impacts on salivary glands’ morphology and the function of salivary glands would also need to be further studied. Investigating sublingual administration (and transdermal route) a with mouse model is difficult. Further experiments in large animals are needed to optimize atropine doses, timing, and the frequency of treatment administration for optimizing the reduction of salivary gland uptake before translation to clinical use. 

Furthermore, the pharmacokinetics of PSMA-based radiotracers need to be considered due to the lower PSMA receptor expression in mice salivary glands when compared to that in humans, and due to the differences between mice and humans in terms of the distribution volume and blood flow in salivary glands [33]. Therefore, the positive effect of pharmacological modulation on early images cannot be extrapolated in terms of dosimetry. The estimation of the absorbed dose needs to integrate key parameters over time using various sophisticated workflows. The next step will consist of evaluating the impact this strategy has on [^177^Lu]Lu-PSMA salivary gland absorbed dose estimates using a dosimetric mouse model.

Our study has multiple limitations. The [^99m^Tc]TcO_4_^−^ or [^68^Ga]Ga-PSMA-11 thyroid uptake could induce a signal spill over to the salivary glands. To reduce this potential bias due to thyroid uptake, the SPECT signal quantification of pertechnetate was limited to the parotid glands, which have more distant from the thyroid than the submandibular glands. 

Finally, combination strategies with synergistic approaches (targeting specific and nonspecific bindings) might offer a great potential for reducing salivary gland exposure, to attempt a clinical impact on salivary gland toxicities. Our preclinical proof-of-concept suggests that the sublingual administration of atropine allows a significant decrease in PSMA-radioligand uptake in the salivary glands and opens new perspectives for salivary-gland-sparing strategies.

## Figures and Tables

**Figure 1 pharmaceutics-14-01276-f001:**
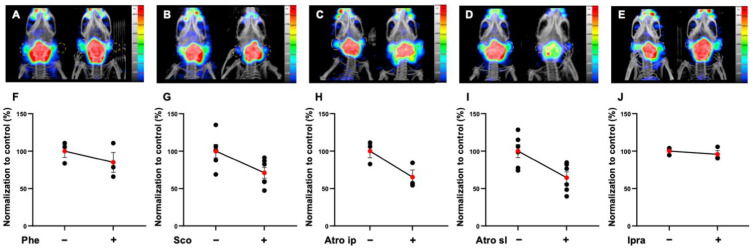
Representative MIP images and normalized quantification of microSPECT/CT biodistribution of [^99m^Tc]TcO_4_^−^ in parotid glands at 45 min after injection in non-tumor-bearing mice premedicated with phenylephrine (Phe) 5 mg kg^−1^ ip (**A**,**F**), scopolamine (Sco) ip 5 mg kg^−1^ (**B**,**G**), atropine (Atro) ip 15 mg kg^−1^ (**C**,**H**), atropine sl 15 mg kg^−1^ (**D**,**I**), or ipratropium sl 4 mg kg^−1^ (**E**,**J**). Red points represent the mean ± SEM; black points represent the sample dispersion. (Statistical test has been performing using paired *t*-test.

**Figure 2 pharmaceutics-14-01276-f002:**
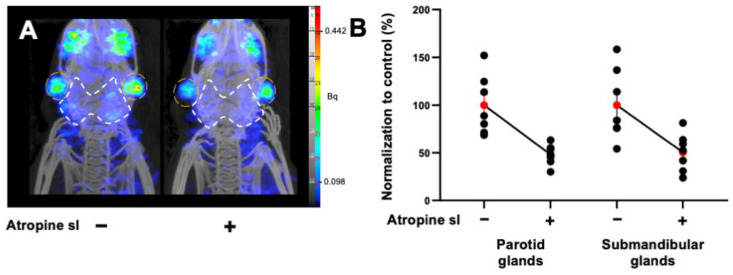
(**A**) Representative MIP images of microPET/CT biodistribution and the quantification of [^68^Ga]Ga-PSMA-11 in the parotids and submandibular glands in non-tumor-bearing mice premedicated with atropine sublingually, at 15 mg kg^−1^. The parotid glands are encircled in white and the submandibular glands are in orange. (**B**) Normalized quantification of [^68^Ga]Ga-PSMA-11 in the parotids and submandibular glands in non-tumor-bearing mice. Red points represent the mean ± SEM; black points represent the sample dispersion.

**Figure 3 pharmaceutics-14-01276-f003:**
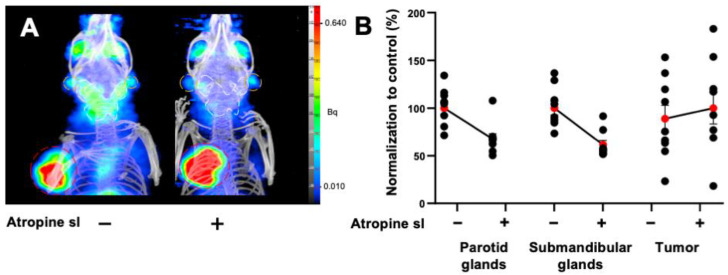
(**A**) Representative MIP images of microPET/CT biodistribution of [^68^Ga]Ga-PSMA-11 in parotid glands and submandibular glands in mice xenografted with prostate adenocarcinoma cells premedicated by atropine sublingually, at 15 mg kg^−1^. The parotid glands are encircled in white, the submandibular glands are in orange, and the tumor is in red. (**B**) Normalized quantification of [^68^Ga]Ga-PSMA-11 in the parotids, submandibular glands, and tumor in prostate-tumor-bearing mice. The red point represents the mean ± SEM; black points represent the sample dispersion.

**Table 1 pharmaceutics-14-01276-t001:** Pharmacological class, molecules, routes, and doses evaluated for the drug screening. Atropine and Ipratropium were sublingually administrated at a fixed dose.

Pharmacology	Drugs	Routes	Dose
α1-adrenergic agonist	Phenylephrine	Intraperitoneal	5 mg kg^−1^
Anti-cholinergic drugs	Scopolamine	Intraperitoneal	5 mg kg^−1^
	Atropine	Intraperitoneal	15 mg kg^−1^
		Sublingual	15 mg kg^−1^ (0.5 mg)
	Ipratropium	Sublingual	4 mg kg^−1^ (0.125 mg)

**Table 2 pharmaceutics-14-01276-t002:** Quantifications of [^99m^Tc]TcO_4_^−^ in the parotid glands at 45 min after injection in non-tumor-bearing mice (ns: not significant; * *p* < 0.05; ** *p* < 0.01).

Pharmacology, Route, Dose	After Treatment (Mean ± SD %ID/cm^3^)	Before Treatment (Mean ± SD %ID/cm^3^)	*p*	Percentage of Variation	*n*
Phenylephrine, intraperitoneal route, 5 mg kg^−1^	24.9 ± 6.8	29.3 ± 4.2	ns	−16.7%	3
Scopolamine, intraperitoneal route, 5 mg kg^−1^	21.1 ± 5.4	29.8 ± 6.5	**	−28.7%	6
Atropine, intraperitoneal route, 15 mg kg^−1^	23.5 ± 6.0	36.0 ± 5.5	*	−34.9%	3
Atropine, sublingual route, 15 mg kg^−1^	24.6 ± 7.3	38.2 ± 8.1	*	−41.6%	6
Ipratropium, sublingual route, 4 mg kg^−1^	21.2 ± 1.2	22.1 ± 1.1	ns	+4.4%	3

**Table 3 pharmaceutics-14-01276-t003:** Quantification of [^68^Ga]Ga-PSMA-11 in the parotids and submandibular glands at 90 min after injection in non-tumor-bearing mice. Statistical tests have been performed by two-way ANOVA followed by a Sidak’s multiple comparison test. *** *p* < 0.001; **** *p* < 0.0001.

Tissue	After Treatment (Mean ± SD %ID/cm^3^)	Before Treatment (Mean ± SD %ID/cm^3^)	*p*	Percentage of Variation	*n*
Parotid glands	0.5 ± 0.1	1.0 ± 0.3	****	−51.6%	7
Submandibular glands	0.3 ± 0.1	0.7 ± 0.3	***	−48.5%	7

**Table 4 pharmaceutics-14-01276-t004:** Quantification of [^68^Ga]Ga-PSMA-11 in the parotids, submandibular glands, and tumor in prostate-tumor-bearing mice. Statistical tests have been performed by two-way ANOVA followed by a Sidak’s multiple comparison test. * *p* < 0.05; **** *p* < 0.0001.

Tissue	After Treatment (Mean ± SD %ID/cm^3^)	Before Treatment (Mean ± SD %ID/cm^3^)	*p*	Percentage of Variation	*n*
Parotid glands	1.1 ± 0.3	1.6 ± 0.3	****	−26.8%	9
Submandibular glands	1.1 ± 0.3	1.8 ± 0.4	****	−37.6%	
Tumor	2.8 ± 1.4	2.5 ± 1.2	*	+17.6%	

## Data Availability

The data presented in this study are available on request from the corresponding author.

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
