# Peer review of "Sublingual Atropine Administration as a Tool to Decrease Salivary Glands’ PSMA-Ligand Uptake: A Preclinical Proof of Concept Study Using [68Ga]Ga-PSMA-11"

_pharmaceutics, 2022, doi:10.3390/pharmaceutics14061276_

Round 1

Reviewer 1 Report

PSMA-targeted radioligand therapy offers a great potency in the treatment of the advanced castration resistant prostate cancer. The substitution of beta emitter nuclides by alpha emitter one can even improve the efficacy of the radiotherapy, however radiotoxicity can hamper the application. Many efforts have been invested to improve the bioavailability of the radioligands, and to decrease the possible side effects. In this work alpha-adrenergic and anticholinergic drugs were applied to save some organs-at-risk, such as parotid- and salivary glands.

Based on my opinion the approach of this issue is really interesting, the research design is simple, but adequate. The research methods – taking account that this is a proof of concept study – are corrects and are interpreted well. Due to the PSMA-based therapy has currently a high interest, the model compound selection is also relevant.

The outcomes are also encouraging. The authors developed a method to decrease significantly the uptake of the pertechnetate and Ga-PSMA in parotid gland and this strategy did not inhibit tumour uptake in xenografted mice. Independently from the positive results, I believe that effective alpha-therapy is impossible without the increasing of radiotracer bioavailability (but this is not a topic of this study).

Obviously this work has some limitations, mainly the difference in the PSMA receptor expression density in mice salivary glands compared to that in humans. Also there are differences between mice and humans blood flow in salivary glands, as it is mentioned by the authors. It is clear the more experiments are needed, but this was a well-established research suitable for publication in the present form.

Author Response

Dear Reviewer 1, 

Sincere thanks to the reviewer for graciously giving his time and his expertise to comment this manuscript.

Best wishes,

Vincent NAIL

Reviewer 2 Report

General Comments:

The paper by Nail et al. attempted to provide a new preventive tool based on premedication of α1-adrenergic and anticholinergic drugs for reducing salivary glands toxicity which represents a side effect of PSMA-targeted radionuclide therapy in patients with advanced castration resistant prostate cancer.

The authors investigated the parotid uptake of [99mTc]pertechnetate ([99mTc]TcO4-) following administration of phenylephrine, atropine, scopolamine or ipratropium by different administration route in mice using SPECT/CT. The most efficient pharmacological treatment was subsequently applied to [68Ga]Ga-PSMA-11 PET/CT in healthy and human prostate adenocarcinoma-xenografted in mice.

The authors found out that scopolamine and atropine decrease the uptake of 99mTcO4- and 68Ga-PSMA-11 in the parotid and salivary glands in mice and concluded that premedication with local administration of atropine could represent an efficient approach for reducing salivary glands uptake in prostate cancer patients after PSMA-targeted radioligand therapy.

Although the topic addresses by the present paper is not particularly new, it remains clinically of interest for the management of patients with advanced prostate cancer who undergo PSMA-targeted radioligand therapy. Alltogether, there are very critical issues in this report that make the manuscript in its present form not suitable for publication in Pharmaceutics. In particular, the presented data based only on the uptake of 99mTcO4- and [68Ga]PSMA-11 in parotid and salivary glands in mice limit conclusions that can be drawn on the potential of sublingual administration of atropine to limit gland toxicity in patients with advanced prostate cancer after treatment with 177Lu-PSMA and 225Ac-PSMA. In addition, a discussion related to the clinical application of atropine in comparison to current preventive methods is lacking or is not convincing. This is particularly important and would increase the scientific and clinical impact of this work significantly.

In details:

  1. The title of the present manuscript is confusing. The authors aimed to develop a new tool to decrease salivary glands PSMA-ligand uptake in patients following treatment with 177Lu-PSMA and 225Ac-PSMA. Once wonders than about the lack of an evaluation of at least the salivary gland uptake of 177Lu-PSMA in this work. This issue has to be addressed and discussed accurately.
  2. Data based exclusively on the evaluation of 99mTcO4- and 68Ga-PSMA-11 in parotid glands in mice limit conclusions that can be drawn on the potential of the sublingual administration of atropine for reducing the uptake of 177Lu-PSMA or 225Ac-PSMA in salivary glands in CRPC patients after radionuclide therapy with 177Lu-PSMA and 225Ac-PSMA.
  3. The contribution or the impact of the evaluation of 99mTcO4- in parotid glands in mice in this study is not clear. It is known that 99mTcO4- is taken up via NIS (mainly into thyroidal tissue or thyroid and breast carcinoma cells), while the molecular target of 177Lu-PSMA and 225Ac-PSMA and other PSMA-ligands is the PSMA receptor, which is only barely expressed in salivary glands in human. This issue needs to be addressed and discussed.
  4. The toxicity of 68Ga-PSMA-11 in salivary glands has not been described clinically at yet. Could the authors discuss this issue more in detail also in context of this study so that the nuclear medicine community could learn more about this established tracer for prostate cancer imaging.
  5. In the experimental section: 99mTcO4- and 68Ga-PSMA-11 were administrated This does not represent the common clinical administration route for 99mTcO4- and PSMA-based ligangs. This is not comprehensible. Could the authors confirm that results obtained by this way are comparable to those obtained following i.v. administration ? The quality of this work could improve significantly if the authors could addressed and discussed this issue. In addition, I would recommend a better description of the in-vivo experiments performed in this work.
  6. The quantification of SPECT signals as well as the data obtained from the quantification of 99mTcO4- and 68Ga-PSMA (PET) uptakes in vivo in parotid glands are in general not convincing.
  7. Accumulation of 99mTcO4- in the parotids following atropine (i.p.) as quantified by ROI technique at 45 minutes post-injection was 0.024 ± 0.004%ID/mm3 vs 036±0.004%ID/mm3 for the control experiment. The local administration using sublingual route of atropine reduced the 99mTcO4- parotids uptake to 0.021 ± 0.005%ID/mm3 compared to 0.034 ± 0.009%ID/mm3 for control condition. In agreement with the study design, atropine by sublingual route was evaluated with 68Ga PSMA-11 PET/CT. At 90 min p.i., the uptake of 68Ga-PSMA-11 in the parotid glands was 0.0005 ± 0.0001%ID/mm3 vs 0.0010 ± 0.0003%ID/mm3 for the control; p < 0.0001). The presented uptake data are really not remarkable. In particular, data from the key evaluation using 68Ga-PSMA-11 PET are very hart to understand and to be accepted as basis for an evaluation of a new preventive tool. This issue need to be discussed more in detail in the discussion section.

Specific remarks: 

  • The English language in the present manuscript is relatively variable. Please cross-check the manuscript for spelling and grammatical errors.
  • The study design and the in vivo PET imaging itself, as well as image analysis have to be described more in details and more accurately in the section “Materials and Methods”.
  • The Discussion part is not always informative. It should improve and the results of the investigation discussed accurately.

Author Response

Response to Reviewer 2 Comments

Sincere thanks to the reviewer for graciously giving his time and his expertise to comment this manuscript.

Please find below, our comments :

Point 1 : The title of the present manuscript is confusing. The authors aimed to develop a new tool to decrease salivary glands PSMA-ligand uptake in patients following treatment with 177Lu-PSMA and 225Ac-PSMA. Once wonders than about the lack of an evaluation of at least the salivary gland uptake of 177Lu-PSMA in this work. This issue has to be addressed and discussed accurately.

Response 1:

The title has been modified according to the reviewer comment.

In the frame of this preclinical proof of concept study, the use of [68Ga]Ga-PSMA-11 PET/CT and [99mTc]TcO4- SPECT appeared more appropriate to us for screening since :

  • the use of imaging radiotracers is easier to implement, simpler to reproduce and at least for PET radiotracer quantification more accurate.
  • Radiotracers ([68Ga]Ga / [99mTc]Tc) using in diagnostic possess shorter half-life than [177Lu]Lu and [225Ac]Ac allowing to perform a before-after evaluation in a short time frame also for practical reason of radioprotection and animal welfare.
  • We think that [68Ga]Ga-PSMA-11 PET/CT imaging and [99mTc]TcO4- SPECT/CT imaging are strong tools to evaluate the impact of the medication on the input function in organs of interest. In this work, the main input function of radiotracers is represented as the perfusion into salivary glands.

As commented by the reviewer, the discussion part has been developed.

Point 2 : Data based exclusively on the evaluation of 99mTcO4- and 68Ga-PSMA-11 in parotid glands in mice limit conclusions that can be drawn on the potential of the sublingual administration of atropine for reducing the uptake of 177Lu-PSMA or 225Ac-PSMA in salivary glands in CRPC patients after radionuclide therapy with 177Lu-PSMA and 225Ac-PSMA.

Response 2 :

We agree that using [68Ga]Ga-PSMA-11 PET/CT imaging and [99mTc]TcO4- SPECT/CT imaging  is not optimal and can’t be totally transposable to PSMA targeted radiotracers using beta-or alpha emitters.

However, atropine (and more largely anti-cholinergic drugs) has had an impact on the input function using two distinct radiotracers with two different uptake mechanisms. Reasonably, we suggest that input function of other PSMA radiotracers could be reduced too but need to be confirmed by future works.

As evoked above, that’s a preclinical preliminary study. The purpose of this work aimed to evaluate the impacts of different molecules on the salivary glands perfusion and tumor. In no way, we wanted to modulate the specific or no specific uptake of PSMA-targeted radiotracers.

As suggested by the reviewer, we implemented and developed the discussion session by this way.

Point 3 : The contribution or the impact of the evaluation of 99mTcO4- in parotid glands in mice in this study is not clear. It is known that 99mTcO4- is taken up via NIS (mainly into thyroidal tissue or thyroid and breast carcinoma cells), while the molecular target of 177Lu-PSMA and 225Ac-PSMA and other PSMA-ligands is the PSMA receptor, which is only barely expressed in salivary glands in human. This issue needs to be addressed and discussed.

Response 3 :

Reviewer is right, but we were not targeting the molecular mechanism – [68Ga]Ga-PSMA-11  and [99mTc]TcO4- have totally different uptake mechanism but by acting on sympathetic and parasympathetic modulation, we aim to module salivary glands perfusion as input function of [68Ga]Ga-PSMA-11  and [99mTc]TcO4-radiotracers.

Modulation of gland perfusion is actually routinely used to manage adverse effect applying cold packs during chemotherapies to reduce alopecia and during PSMA-targeted RLT to manage xerostomia (Rugo and Voigt, 2018; van Kalmthout et al., 2018) .

On the other hand, [99mTc]TcO4- SPECT/CT imaging is used as a tool to evaluate the effect of the different approved drugs on the input function of salivary glands. [99mTc]TcO4- SPECT/CT imaging appeared to be the most practical for screening as described in the point 1.

That’s why, once a satisfactory strategy has been identified using, we decided to evaluate it with compatible route for human use (sublingual route) and then using [68Ga]Ga-PSMA-11 nearest to our objectives. Next step would be to evaluate PSMA-based radiotracers for RLT.

The starting point/paradigm are that a reduction of the salivary glands uptake, may reduce potential adverse effects, even if, there is no clear relation between PSMA uptake – adverse effects – and impact on quality of life.

This issue has been implemented in discussion section.

Point 4 : The toxicity of 68Ga-PSMA-11 in salivary glands has not been described clinically at yet. Could the authors discuss this issue more in detail also in context of this study so that the nuclear medicine community could learn more about this established tracer for prostate cancer imaging.

Response 4:

As reviewer evoked, [68Ga]Ga-PSMA-11 has “no adverse effect” in literature even in our department.

As described above, we tried to target input function [68Ga]Ga-PSMA-11 for purpose to translate it further with PSMA targeted RLT radiotracer.

Point 5 : In the experimental section: 99mTcO4- and 68Ga-PSMA-11 were administrated This does not represent the common clinical administration route for 99mTcO4- and PSMA-based ligangs. This is not comprehensible. Could the authors confirm that results obtained by this way are comparable to those obtained following i.v. administration ? The quality of this work could improve significantly if the authors could address and discussed this issue. In addition, I would recommend a better description of the in-vivo experiments performed in this work.

Response 5:

Intraperitoneal route has been preferred because:

  • IP administration in mice is characterised by a slow and delayed blood input peak compared to direct IV route. IP route seemed to us more appropriate to mimic RLT administration than IV bolus administration (Kim et al., 2011).

As evoked by the reviewer, the discussion section has been implemented with these elements.

Point 6 : The quantification of SPECT signals as well as the data obtained from the quantification of 99mTcO4- and 68Ga-PSMA (PET) uptakes in vivo in parotid glands are in general not convincing.

Response 6:

We added complement information in Material & methods part.

To improve robustness of our quantification, cameras PET/CT and SPECT/CT are annually calibrated and updated by manufacturers. Additionally, our cameras are internally calibrated every month.

Regarding the quantifications, evaluations have been independently performed by two different experimenters.

Point 7 : Accumulation of 99mTcO4- in the parotids following atropine (i.p.) as quantified by ROI technique at 45 minutes post-injection was 0.024 ± 0.004%ID/mm3 vs 036±0.004%ID/mm3 for the control experiment. The local administration using sublingual route of atropine reduced the 99mTcO4- parotids uptake to 0.021 ± 0.005%ID/mm3 compared to 0.034 ± 0.009%ID/mm3 for control condition. In agreement with the study design, atropine by sublingual route was evaluated with 68Ga PSMA-11 PET/CT. At 90 min p.i., the uptake of 68Ga-PSMA-11 in the parotid glands was 0.0005 ± 0.0001%ID/mm3 vs 0.0010 ± 0.0003%ID/mm3 for the control; p < 0.0001). The presented uptake data are really not remarkable. In particular, data from the key evaluation using 68Ga-PSMA-11 PET are very hart to understand and to be accepted as basis for an evaluation of a new preventive tool. This issue need to be discussed more in detail in the discussion section.

Response 7:

Respectfully to international nomenclature guidelines for molecular imaging studies, Results are expressed as  %ID/mm3, that resulting in decimal values, however salivary gland uptake values in this present study are comparable to those already reported in the literature even for [68Ga]Ga-PSMA-11  and [99mTc]TcO4-experiments

Other difference could be explained by the potential impact of specific activity of PSMA based radiotracer or and the inter-variability between the animal strains

We converted from %ID/mm3 to %ID/cm3 to have unit similar to %ID/g. For more details and clarity, absolute quantification has been summarized in different tables (2,3,4)

Specific remarks: 

  • The English language in the present manuscript is relatively variable. Please cross-check the manuscript for spelling and grammatical errors

The manuscript has been reviewed by English native

  • The study design and the in vivo PET imaging itself, as well as image analysis have to be described more in details and more accurately in the section “Materials and Methods”.

We add some information according to your comment in M&M section

  • The Discussion part is not always informative. It should improve and the results of the investigation discussed accurately.

We completed and adjusted the discussion section

Thank you in advance for your comments and I'm looking forward to your answers, 

Best wishes,

Vincent NAIL

Reviewer 3 Report

This paper presents the  preclinical results of the administration of atropine  in the salivary glands  and tumour uptake of [68Ga]Ga-PSMA-11 in mice

The study also uses [99mTc]TcO4- microSPECT/CT as screening method with a group that modulate  the autonomic innervation of salivary glands.

Although the objective to reduce the salivary uptake of [68Ga]Ga-PSMA-11 in salivary glands is important in order to improve the clinical results of the therapy using analogous therapeutic radiopharmaceuticals, this study has in my opinion many limitations:

Justification of the use of pertechnetate as screening agent is necessary since uptake of PSMA in salivary glands does not occur by the same mechanism than uptake of pertechnetate

Justification of the selected administration routes for the different pharmaceuticals in the animal model is also required.

In case of the intraperitoneal administration, which is the correlation of  this administration route with the use in humans?

Regarding the expression of the results:

the graphs showing uptake values are very small. Inclusion of a Table with the numerical results would be more informative.

the uptake values  and standard deviations are very small . The calculation based on manually drawn regions of interest is enough  to obtain these small values with sufficient precision?

Regarding the uptake in tumour following administration of atropine the authors claim that "Notably, the tumor [68Ga]Ga-PSMA-11 PET signal was increased by 18% following sublingual atropine compared to control (respectively 0.0035 ± 0.0015%ID/mm3 vs 0.0030 ± 0.0001%ID/mm3; p = 0.012; PV = 17.6%; n = 9) "  However the value of 0.0030 is included in the range 0.0035 ± 0.0015. These values seem to be the same to me.

There is also a small inconsistency with the uptake increase in tumour. In page 6 line 217 it says 18% increase and in the discussion the increase is mentioned as 27%.

Author Response

Response to Reviewer 3 comments

Sincere thanks to the reviewer for graciously giving his time and his expertise to comment this manuscript.

Please find below, our comments :

Point 1 : Justification of the use of pertechnetate as screening agent is necessary since uptake of PSMA in salivary glands does not occur by the same mechanism than uptake of pertechnetate :

Response 1 :

Reviewer is right, but we were not targeting the molecular mechanism – [68Ga]Ga-PSMA-11  and [99mTc]TcO4- have totally different uptake mechanism but by acting on sympathetic and parasympathetic modulation, we aim to module salivary glands perfusion as input function of [68Ga]Ga-PSMA-11  and [99mTc]TcO4-radiotracers.

Modulation of gland perfusion is actually routinely used to manage adverse effect applying cold packs during chemotherapies to reduce alopecia and during PSMA-targeted RLT to manage xerostomia (Rugo and Voigt, 2018; van Kalmthout et al., 2018).

On the other hand, [99mTc]TcO4- SPECT/CT imaging is used as a tool to evaluate the effect of the different approved drugs on the input function of salivary glands. This modality of imaging using [99mTc]TcO4- appeared to be the most practical for screening

That’s why, once a satisfactory strategy has been identified, we decided to evaluate it with compatible route for human use (sublingual route) and then using [68Ga]Ga-PSMA-11 nearest to our objectives. Next step would be to evaluate PSMA-based radiotracers for RLT.

The starting point/paradigm are that a reduction of the salivary glands’ uptake, may reduce potential adverse effects, even if, until now, there is no clear relation between uptake – adverse effects – and impact on quality of life.

This issue has been implemented in discussion section.

Point 2 : Justification of the selected administration routes for the different pharmaceuticals in the animal model is also required

Response 2 :

Intraperitoneal route has been preferred because:

  • IP administration in mice is characterised by a slow and delayed blood input peak compared to direct IV route. IP route seemed to us more appropriate to mimic RLT administration than IV bolus administration (Kim et al., 2011).

As evoked by the reviewer, the discussion section has been implemented with these elements

Point 3 : In case of the intraperitoneal administration, which is the correlation of  this administration route with the use in humans?

Response 3 :

As described above, IP administration has been chosen for the practical aspects to perform screening and its profile of absorption.

The final objective was to translate to a compatible route for human use, that why, we studied sublingual route.

Point 4 : Regarding the expression of the results: the graphs showing uptake values are very small. Inclusion of a Table with the numerical results would be more informative

Response 4:

Respectfully to international nomenclature guidelines for molecular imaging studies, Results are expressed as  %ID/mm3, that resulting in decimal values, however salivary gland uptake values in this present study are comparable to those already reported in the literature even for [68Ga]Ga-PSMA-11  and [99mTc]TcO4-experiments

Other difference could be explained by the potential impact of specific activity of PSMA based radiotracer or and the inter-variability between the animal strains

We converted from %ID/mm3 to %ID/cm3 to have unit similar to %ID/g. For more details and clarity, absolute quantification has been summarized in different tables (2,3,4)

 We modified the “results” section and did a table according to the reviewer’s comment

Point 5 : the uptake values and standard deviations are very small. The calculation based on manually drawn regions of interest is enough  to obtain these small values with sufficient precision?

Response 5  :

Performed manually regions of interest appeared to be the best method :

  • Automatic segmentation method is not appropriate due to the studied organs and the anatomical inter-variabilities between animals

To reinforce the results, evaluations/quantifications have been independently performed by two different experimenters for PSMA radiotracer biodistribution.

As evoked above, we converted from %ID/mm3 to %ID/cm3 to have unit similar to %ID/g.

Point 6  : Regarding the uptake in tumour following administration of atropine the authors claim that "Notably, the tumor [68Ga]Ga-PSMA-11 PET signal was increased by 18% following sublingual atropine compared to control (respectively 0.0035 ± 0.0015%ID/mm3 vs 0.0030 ± 0.0001%ID/mm3; p = 0.012; PV = 17.6%; n = 9) "  However the value of 0.0030 is included in the range 0.0035 ± 0.0015. These values seem to be the same to me.

Response 6 : Indeed, standard deviations intersect, the test is still significant according to the used statistical test and the number of animals explored in this condition.

Point 7 : There is also a small inconsistency with the uptake increase in tumour. In page 6 line 217 it says 18% increase and in the discussion the increase is mentioned as 27%.

Response 7 :The sentence has been modified.

Thank you in advance for your comments and I'm looking forward to your answers, 

Best wishes,

Vincent NAIL

Round 2

Reviewer 2 Report

I have re-evaluated the revised manuscript "Sublingual atropine administration as a tool to decrease salivary glands PSMA-ligand uptake: a preclinical proof of concept study using [68Ga]Ga-PSMA-11" (MS ID: pharmaceutics-1719812) by Vincent Nail, Beatrice Louis, Anais Moyon, Adrien Chabert, Laure Balasse, Samantha Fernandez, Guillaume Hache, Philippe Garrigue, David Taïeb, and Benjamin Guillet as well as the provided point-by-point responses to my initial review.

The authors addressed all concerns and suggestions raised at the time of the first review and the revision has sufficiently strengthened the manuscript to deserve publication in the journal “Pharmaceutics”. Therefore, I recommend the manuscript in its present form for publication in Pharmaceutics.

Reviewer 3 Report

Author´s responses and changes made in the text are sufficient to allow publication in the present form